# Understanding the Ecosystem Carrying Capacity for *Romanichthys valsanicola*, a Critically Endangered Freshwater Fish Endemic to Romania, with Considerations upon Trophic Offer and Behavioral Density

Laurențiu Burlacu * , Gyorgy Deak *, Mădălina Boboc *, Marius Raischi, Elena Holban, Isabela Sadîca and Abdulhusein Jawdhari

National Institute for Research and Development in Environmental Protection, 294 Splaiul Independenței Blv, District 6, 060031 Bucharest, Romania; raischimarius@yahoo.com (M.R.); holban.elena@yahoo.com (E.H.); isabela.sadica@incdpm.ro (I.S.); abdul.jawdhari@incdpm.ro (A.J.)
* Correspondence: laurentiu.burlacu@incdpm.ro (L.B.); dkrcontrol@yahoo.com or incdpm@incdpm.ro (G.D.); madalina.boboc@incdpm.org (M.B.)

**Abstract:** The most recent assessment (2008) of the IUCN Red List of Threatened Species classified *Romanichthys valsanicola* as critically endangered (CR). In December 2022, an expert team from the National Institute for Research and Development in Environmental Protection Bucharest investigated the presence of the species in historical locations and in other potential sites. The authors correlated public data to the actual habitat area to calculate the potential species density in relation to the specific territorial behavior. The ecological carrying capacity was represented by the consensus between the behavioral density limitations and the trophic limitations of the actual species potential habitat. Both trophic availability and populational density present encouragingly high values for the sculpin perch in the Valsan River, indicating that the natural habitat could host a considerably higher number of individuals.

**Keywords:** ecosystem carrying capacity; maximum supported population; food availability; *Romanichthys valsanicola*; Valsan River; IUCN critically endangered (CR)

## 1. Introduction

*Romanichthys valsanicola* M. Dumitrescu, P. Bănărescu & N. Stoica, 1957, the Romanian darter, sculpin perch or asprete, as it is commonly known, is one of the 52 most threatened continental inland freshwater fish species in Europe [1], justifying the increased academic interest for this species and its inclusion within significant legal frameworks and conservation-related documents at national, regional, community and global levels [2–4]. Discovered in 1956, 67 years ago, and described and published in 1957 [5] by P. Bănărescu, N. Stoica and M. Dumitrescu, the species is currently listed as critically endangered (CR) according to the last assessment (2008) [6] of the IUCN Red List of Threatened Species after being previously assessed as endangered (EN) from 1986 to 1994, and it was subsequently moved into a higher conservation concern category [6] in 1996.

The sculpin perch is the smallest representative of the Romanichthyni tribe with a distribution restricted to the Danube basin, where *Romanichthys valsanicola* is endemic to the Valsan River, a tributary of the Arges River, located in the Arges County in the central Subcarpathian area of Romania south of the Fagaras Massif (Figure 1). The *Zingel* species populate the Danube, Rhone and Vardar systems [7], having a less localized distribution. The tribe includes a total of five species, namely *Romanichthys valsanicola* (Figure 2) and the four species [8–11] within the sister genus *Zingel* [11–13]. The number of species in the Romanichthyni tribe may differ, since Banarescu [8] initially considered *Z. balcanicus* as a subspecies of *Z. zingel* and not a valid species, as more recent studies indicate [9,10,14].

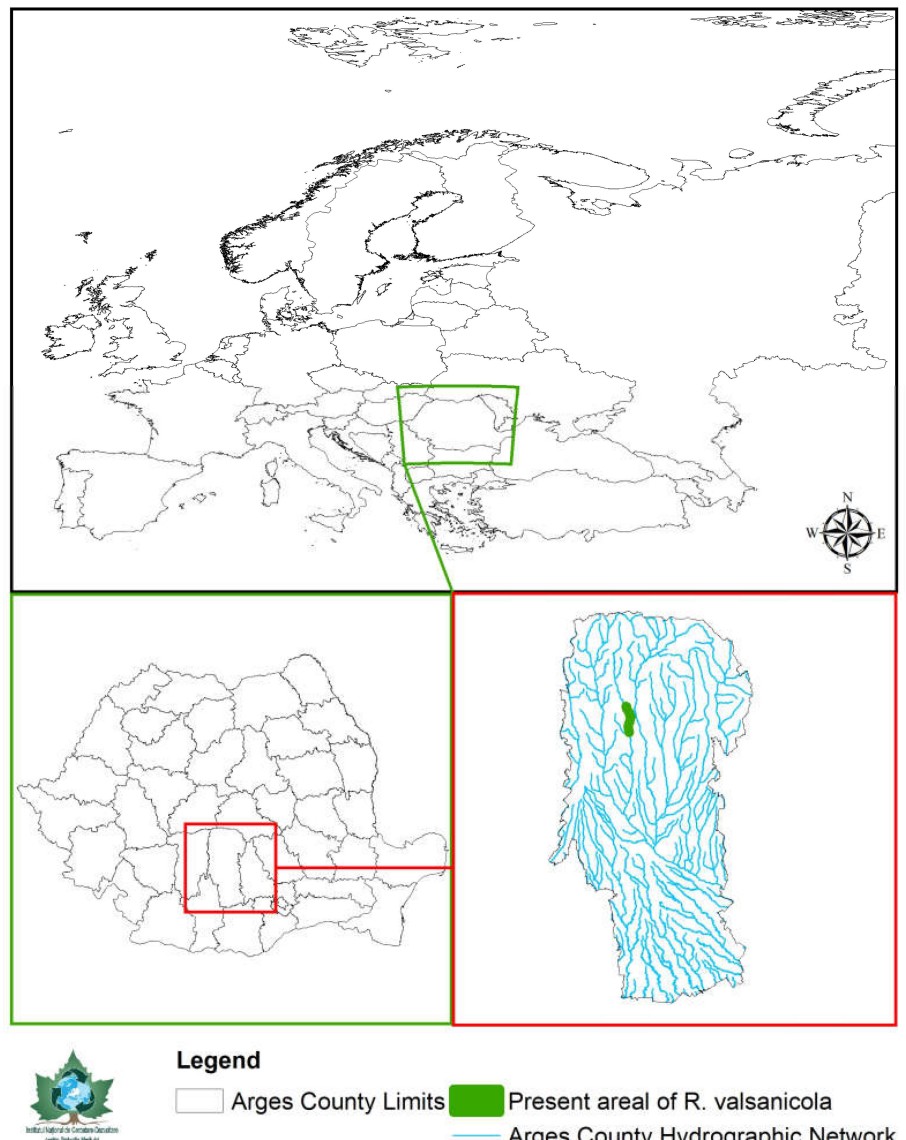

**Figure 1.** Localization of the Valsan River and the present *Romanichthys valsanicola* areal within the Argeş County limits and the Romanian borders.

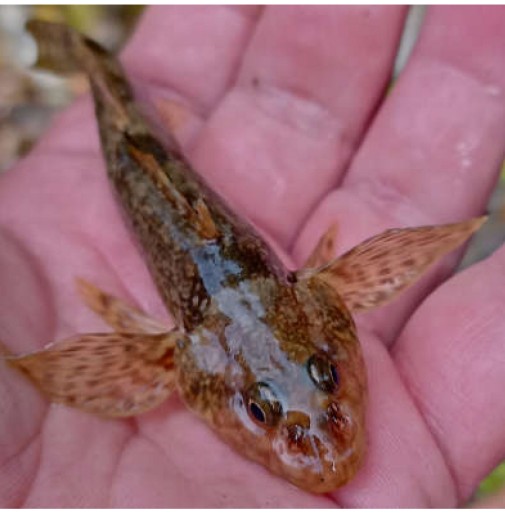

**Figure 2.** *Romanichthys valsanicola* during the field investigation activities.

The tribe illustrates distinct adaptations to the particular fast-flowing highland aquatic stream habitats (for *Romanichthys*) or larger flowing water courses (*Zingel*), such as a fusiform, elongated body shape adapted to resist downstream displacement, a downwards-oriented mouth and a trophic spectrum consisting mainly of rheophilic invertebrates (Ord. Ephemeroptera, Plecoptera, Trichoptera) [15–18]. However, Stepien and Haponski [11] argue in Chapter 1, "Taxonomy, Distribution, and Evolution of the Percidae—Biology and Culture of Percid Fishes Principles and Practices", that *Romanichthys* might not be a valid genus but part of the genus *Zingel*, as suggested by mtDNA evidence in the respective study [11], since prior DNA studies [19–21] did not include *Z. asper* and *Z. balcanicus*. The scuplin perch, a nocturnal, bottom-dwelling species [6], is characterized by an elongated body with a large head, similar to *Cottus gobio*, a more common species with which the sculpin perch shares the natural habitats in the Valsan River. Other distinctive characteristics are the clearly separated dorsal fin, where the second dorsal exceeds the first dorsal in terms of height, the eyes positioned on the dorsal side of the head, the breeding tubercules and the fine opercular serrations [11,13,22] as well as the absence of a swimbladder [11].

Although a rather novel species among the freshwater fish taxa, *Romanichthys valsanicola*, the only representative of a monospecific genus in the Romanichthyini tribe Dumitrescu, Bănărescu and Stoica 1957 [7,11,23], is considered by some authors as a relict taxon [24,25], with the tribe Romanichthyni having diverged from *Sander* 24.6 million years ago (most likely between 11.5 and 39.6 million years ago) [11,26], in the late Oligocene [21], as a probable consequence of tectonic activity [11,27]. The natural areal of the sculpin perch covered, historically, the Argeş River and two of its tributaries (Valsan and Raul Doamnei) [28], yet the last IUCN Red List Report [6] indicates that the species survives only in a 1 km stretch of the Valsan River, and it argues that the drastic populational decline is linked to the construction of a water reservoir on the River Arges in 1965, when no water was left in the river downstream of the dam.

The most frequently indicated threats for the species are represented by illegal stone extraction for construction purposes, overfishing, which is sometimes claimed to be associated with the growing interest of the scientific community for this novel species, and damming of the river, which influenced the flow regime and probably the structure and composition of the aquatic invertebrate fauna that makes up the food of the sculpin perch, although this aspect is disputed by some authors [29]. Other causes for the populational decline of this species are represented by mining and quarrying, especially by rock extraction from the riverbed, the development of transport infrastructure (roads and railroads), logging and wood harvesting, dams and water management/use, the use of pesticides and fertilizers, and organic/inorganic waste depositing within the river bed [6,24,25,28,30–32].

Up to present times, except for a peer-reviewed publication dealing with the food and feeding habits of *Romanichthys valsanicola* (Găldean et al. 1997 [18]) that focused on identifying food items found in the stomach contents of 34 specimens preserved in the collections of the Institute of Biology, Bucharest, no studies were carried out regarding the carrying capacity for this species based on food availability. According to the authors, the food spectrum of the sculpin perch consists mainly of the aquatic larvae of insects of the orders Ephemeroptera (67%), Plecoptera (10.9%) and Trichoptera (7%), with other taxa such as Blephariceridae (10%) and Chironomidae (5.4%) being also found. Some Oligochaeta and Gammaridae appear only sporadically and with non-significant values [18]. Due to the lack of data concerning the spatial distribution and biomass abundance of the Blephariceridae in the Valsan River, the respective data could not be used in the present study, and only the values of the orders Ephemeroptera, Trichoptera and Plecoptera, as well as for the Chironomidae family were employed, summing up to a total of over 90% of the food items identified.

The present study attempts to correlate the prior trophic spectrum data with the food availability and the territorial/behavioral thresholds of the species and to provide an estimation of the specific ecological carrying capacity within the natural areal of *R. valsanicola*.

The conservation efforts that aimed at preventing the extinction of *R. valsanicola* led to a EU Life program [33] that addressed some natural habitat reconstruction and preser-

vation issues, as well as an ex situ conservation attempt [34], throughout a collaboration partnership within the same program where seven specimens were taken to the Institute for Animal Ecology, Federal Agency for Nature Conservation, in Bonn, Germany. Although the attempt failed and the individuals died, the experiment provided valuable data concerning the reproduction biology and habits as well as a starting point for future ex-situ reproduction initiatives.

## 2. Materials and Methods

### 2.1. Specific Areal

The previously known distribution areal of the species was documented throughout a literature review [33,35], and all publicly available research papers referring to the topic were taken into account. A field trip took place in December 2022, when the expert team from the INCDPM (National Institute for Research and Development in Environmental Protection) investigated the presence of the species in the historical locations and in other potential sites. A number of 47 *Romanichthys valsanicola* individuals were captured, employing electrofishing devices (Hans Grassl L60 II HI), and they were subsequently returned to their natural environment after species identification and measurements (total length in mm [36], standard length in mm [37], weight in g) sampling were carried out.

The corresponding area of the Valsan River sector where *R. valsanicola* identifications occurred was digitized (ESRI ArcGIS 10.8), in a WGS 84 projection, employing Esri, Maxar, Earthstar Geographics, and the GIS User Community spatial imagery sources (ESRI ArcGIS servers). The spatial features were measured, and the output values were employed in the subsequent analyses and forecasts as well as for comparation purpose with the historical occurrence data.

### 2.2. Home Range

The specific home range, seen here as the size of the maximum individual territory, within which a specimen explores, feeds, forages and interacts with other intra or inter-specific individuals (and differs considerably in terms of size from the individual territory concept, represented by a considerably smaller occupied area where aggressive responses occur when trespassed by conspecific individuals) is a critical metric in selecting optimal population sizes, since most authors discussing the sculpin perch behavior make references to its intra-specific aggressivity and territorial traits [28,31,33,35,38]. The authors correlated the available public data to the actual *Romanichthys valsanicola* habitat area to calculate the potential species density in relation to the specific territorial behavior. The values were used as threshold conditions to be applied to the trophic carrying capacity calculations.

### 2.3. Food Composition

All stomach content samples analyzed in Găldean et al., 1997 [18] were taken into account in the calculations for fullness index and average weight of daily food intake. The number of preys per order found in the study was divided by the number of stomach contents analyzed to provide a mean value for preys per order per stomach content, and the result was multiplied by the average weight of a larva belonging to that order to calculate the mean individual fulness index. Larvae weights were documented from the specialty literature by dividing the value of sampled biomass per order with the number of items per sample [29].

### 2.4. Food Availability

Food availability estimates took into account the research of Vlăduţu 2003 [29], who investigated the presence in the Valsan River as well as the qualitative and quantitative composition of aquatic invertebrate species that make up the preferred prey species for *R. valsanicola*. The estimated values were expressed as $g/m^{-2}$, and they reflected the frequency and abundance of the respective Ephemeroptera, Trichoptera and Plecoptera larvae observed during a field study in 2003. Since the study [29] addressed a river

sector larger than the actual areal populated by the asprete, only the sampling stations within the area of interest were employed in the calculations concerning our study. The trophic availability was calculated as the product of biomass abundance per order (from all sampling stations within the actual area of the potential habitat) and the actual area of the respective optimal habitat where the sculpin perch was detected. The biomass availability per order was finally summed up for all orders (Ephemeroptera, Plecoptera and Trichoptera) and further on multiplied by 0.85 to reflect the real contribution of the analyzed aquatic invertebrate orders to the sculpin perch diet (85%, as reflected by the studies of Galdean et al. [18]). The resulted value is rather idealistic, since it assumes that *Romanichthys valsanicola* does not share the trophic niche with other species. As a correction factor, a value of 4.8% *R. valsanicola* representativity in the native fish assemblages was employed, according to our field investigations as well as previous studies [30,35]. Nevertheless, since most of previous studies employed solely numeric records (number of individuals per species) and did not provide ponderal values for captures (biomass per species), further corrections and thorough investigations are required for more reliable estimates, having in mind the small weight of the sculpin perch compared to other fish species found in the same habitats.

### 2.5. Ecological Carrying Capacity

In order to calculate the ecological carrying capacity for the species in the actual habitat, both results from the density-dependent (home range) and the trophic offer estimates were taken into account. Estimates for the biomass percent occupied by *Romanichthys valsanicola* were calculated from our field observations in agreement with the available literature sources.

## 3. Results and Discussion

The current distribution areal was evaluated as stretching along more than 10 km according to recent public information [39] and personal unpublished data, indicating a larger probable habitat dimension than the historical findings. Digitization of the available spatial data concerning the diachronic distribution of *R. valsanicola* (Figure 3) indicates that there is a most probable areal expansion of approximately 5–6 km of river, representing 54.55% when compared to the historical data [30,33,35], which indicate various contradicting values, ranging from 1 km [24,28], 5 km [30] to 7–9 km [33,35] (Table 1).

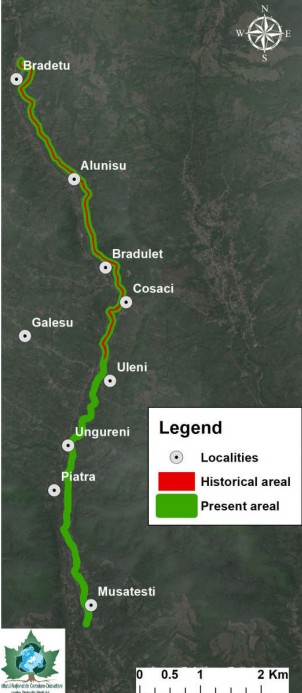

**Figure 3.** Distribution of the historical and actual habitat of *Romanichthys valsanicola*.

**Table 1.** Actual river stretch length populated by *Romanichthys valsanicola* compared to the historical data.

| Reference for (Period) | Historical (km) | Actual (km) | "New" Length of River (km) | Variation Compared to Historical Data (%) |
|---|---|---|---|---|
| Vlăduţu 2013; Bănărescu 1995; (1992–1993) | 1 | 11 | 10 | 90.91 |
| Truţă 2016; (2016) | 5 | 11 | 6 | 54.55 |
| Ionaşcu 2009; (2009) | 7–9 | 11 | 4–2 | 36.36–18.18 |
| Bănărescu 1995; (1956–1965) | 21 | 11 | −10 | −90.91 |

In terms of habitat area, this translates as a 49% increment (Table 2). Nevertheless, this aspect should not be taken as an areal expansion *sensu stricto* but rather as the effect of increased research interest and more accurate methodologies being employed.

**Table 2.** *Romanichthys valsanicola* actual habitat area compared to the average of historical values.

| Average Historical Area (Ha) | Present Area (Ha) | "New" Area (Ha) | Percent Increase Compared to Historical Data (%) |
|---|---|---|---|
| 6.3 | 12.36 | 6.06 | 49.03 |

*3.1. Behavioral Carrying Capacity*

As a main, fundamental prerequisite, we need to differentiate between the concepts of home range and territory. Several studies [40–43] distinguish between home range and territory. Home range is seen as the area over which an individual carries out its feeding, foraging and explorative activities [40] with lax borders that may overlap with those of inter- or intraspecific individuals. Territory is usually of smaller size and seen as a core of the home range with strict borders, where the individual response toward trespassing would in most cases be represented by aggressivity [41]. In this respect, the authors have discarded the home range values presented by some authors [35] and focused on the territory estimates. Yet, Ionaşcu 2009 [44] argues that the natural territory, as observed during the field investigations, would be about 2–3 m$^2$, while in the deeper sections of the Valsan River, no more than two to three individuals were captured (in 1992–1994), suggesting that the territory size could be about 10–15 m$^2$. We calculated the average of the extreme values as a consensus and agreed on an individual territory size of 8.5 m$^2$ to represent the mean of the territory area value. In this context, for an actual area of potential habitat of 12.36 hectares, the behavioral density thresholds for *Romanichthys valsanicola* in the Valsan river would range between 7 and 50 individuals per 100 m$^2$ and with potential population sizes between 8241 and 61,804 individuals, according to the potential variation of the territory size, as presented in Table 3. Further studies are required to identify more representative data for the population densities of *Romanichthys valsanicola* in its natural habitat, since previous studies indicate that the level of aggressiveness tends to decrease if the environment offers a large and varied number of riverbed items such as large stones or boulders, which probably play an important role in limiting the visibility and interaction of territorial individuals, playing thus an important role in increasing the potential behavioral density.

**Table 3.** Estimations of potential individual densities.

| Territory Size Descriptive Statistics | Value of Territory Size (m$^2$) | Actual Potential Habitat Area (m$^2$) | Potential Number of Individuals According to the Territorial Limitations | Individual Density Per 100 m$^2$ |
|---|---|---|---|---|
| Minimum | 2 | 123,608.1 | 61,804 | 50 |
| Average | 8.5 | 123,608.1 | 14,542 | 12 |
| Maximum | 15 | 123,608.1 | 8241 | 7 |

*3.2. Trophic Carrying Capacity*

The local fish assemblage structure (Table 4) sums up a number of 16 species and indicates that the ichthyofauna of the Valsan River is dominated by Barbus petenyi and Squalius cephalus with biomass representations of 55.4% and 23.2%, respectively, followed by Salmo trutta, with 10.1% of the total biomass, while the rest of the species have significantly lower representations. *Romanichthys valsanicola* shares a value of 4.7% of the total fish biomass in its natural habitat. This calculated percentual biomass participation agrees partially with the findings of Truţă and Stancu, 2016 [30], who found 1 sculpin perch individual in a sample size of 27 fishes belonging to 5 species, but they do not offer any information concerning the total specific biomass of the sample.

**Table 4.** Fish assemblages in the Valsan River and percentual biomass representation per species (ordered by total weight, maximum to minimum).

| Species | Total Weight (g) of Captured Individuals Per Species | Percentual Biomass Representation in the Fish Assemblage (%) |
|---|---|---|
| *Barbus petenyi* | 5465 | 55 |
| *Squalius cephalus* | 2283 | 23 |
| *Salmo trutta* | 1002 | 10 |
| *Romanichthys valsanicola* | 470.1 | 4 |
| *Cottus gobio* | 230.2 | 2 |
| *Sabanejewia balcanica* | 99 | 1 |
| *Sabanejewia romanica* | 58.7 | 0.6 |
| *Phoxinus phoxinus* | 53.8 | 0.55 |
| *Romanogobio uranoscopus* | 38.1 | 0.39 |
| *Alburnus alburnus* | 33 | 0.33 |
| *Alburnoides bipunctatus* | 25 | 0.25 |
| *Barbatula barbatula* | 21.8 | 0.22 |
| *Salvelinus fontinalis* | 17 | 0.17 |
| *Barbus barbus* | 11.2 | 0.11 |
| *Pseudorasbora parva* | 8 | 0.08 |
| *Gobio gobio* | 3.6 | 0.04 |

Yearly food demand per individual (Table 5) was calculated by corroborating data from Gâldean, 1997 [18] and Vlăduțu, 2003 [29], regarding the cumulated weight of prey items per order, per individual, in order to obtain the average daily food intake by individual. Data from Gâldean, 1997 [18] offered information about the average number of prey per order, taking into account all the 35 stomach contents analyzed by the authors. Since authorities for the sculpin perch indicate that the species has the most primitive and short digestive tract [5,18] of the percid series, indicating thus that the digestion processes may have a fast rate for this species, hence, we can assume that the fullness index value represents the daily food intake per individual. The corroborated data of Vlăduțu 2003 [29] and Galdean [18] were used to estimate the average weight of prey items per order, which was multiplied by the average number of prey belonging to the respective order found in each stomach content. The resulted value represents the daily individual food intake, which is expressed as the fullness index [45–48]:

$$\text{FI} = \text{FW}/\text{W} \times 100$$

where:

FI = fullness index (% of gut contents weight from the fish weight);
FW = food weight identified within the stomach content (g);
W = body weight of each sampled individual.

The FW was extrapolated for a whole year length to calculate the yearly food demand per individual. The reference individual weight was estimated from the measurements carried out by the study team during the field investigations. Although it can be argued that trophic activity is not constant during the whole year, since the metabolic rate of *Romanichthys valsanicola* may be slowed within the cold periods of the year, in the absence of fullness index calculations in relation to environmental temperatures, we assumed a constant food intake and a constant environmental food productivity over the year.

**Table 5.** Yearly food demand per individual and intermediary calculation steps.

| Order | Ephemeroptera | Plecoptera | Trichoptera |
|---|---|---|---|
| Average number of preys | 10.7429 | 1.8000 | 1.0857 |
| Average weight of prey item/order (g) | 0.0012 | 0.0012 | 0.0270 |
| Total weight of preys/order (g) | 0.0133 | 0.0021 | 0.0293 |
| Average individual stomach content weight (g) | 0.05 | | |
| Average individual weight (g) | 9.99 | | |
| Average individual fullness index (daily food demand as % of individual body weight) | 0.46 | | |
| Average yearly food requirements/individual | 16.78 | | |

The trophic offer in the Valsan River was founded on the findings of Vlăduțu, 2003 [29], where the wet weight values per order were cumulated from all sampled stations, and a monthly average was computed and further on multiplied by 12 to provide the yearly food productivity/m$^2$. This value was then multiplied by the area of the actual habitat where *Romanichthys valsanicola* occurrences were documented, returning the yearly ecosystem carrying capacity for the sculpin perch. The processed data presented in Table 6 illustrate that the Valsan River, despite its documented degradation in habitat condition, still offers a remarkable trophic support for the sculpin perch, being able to provide a trophic base for more than 100,000 individuals, at its maximum occupancy. These findings agree with those of Vlăduțu, 2003 [29], who underlines that the aquatic invertebrate fauna that constitute the trophic base for *Romanichthys valsanicola* has increased in both richness and abundance compared to previous studies (Vlăduțu 2003 citing TATOLE, V (1993): Noi considerații asupra situației critice a endemitului *Romanichthys valsanicola*—Ocrot. nat. med. înconj., tom 37, București: 125–127), and it invalidates the hypothesis that the sculpin perch population is decreasing due to a strong decrease in the food offer. In this context, the actual reasons for the species' decline should be looked for elsewhere, and thorough population size and structure investigations need to be carried out, including mark-recapture studies, to document a realistic foundation for various metrics of interest, such as actual individual numbers per age class and sex category, territory size, specific feeding seasonality and ecosystem trophic productivity.

### 3.3. Ecological Carrying Capacity

The final step of the present study was represented by the consensus between the behavioral (territorial aggressivity) density limitations and the trophic limitations of the actual *Romanichthys valsanicola* potential habitat by choosing whichever value is lower and deciding which of the indicators limit the thresholds of the sculpin perch in its natural habitat. The results are illustrated in Table 7.

**Table 6.** Trophic carrying capacity for *Romanichthys valsanicola*—calculations for the potential habitat area within the Valsan River.

| | Productivity (g/m$^2$) |
|---|---|
| Ephemeroptera (g/m$^2$) | 7.05 |
| Plecoptera (g/m$^2$) | 3.00 |
| Trichoptera (g/m$^2$) | 15.51 |
| Chironomidae (g/m$^2$) | 3.35 |
| Total/Month (g/m$^2$) | 28.91 |
| Total/Year (g/m$^2$) | 346.95 |
| Potential habitat area (m$^2$) | 123,608.11 |
| Total habitat productivity/year (g) | 42,886,327.31 |
| Average yearly food requirements/individual (g) | 16.78 |
| Ideal yearly trophic carrying capacity (individuals represent 100% of the fish assemblage) | 2,555,800 |
| Specific representativity within the local fish assemblages (*Romanichthys valsanicola* % biomass within the local fish assemblages) | 0.0477 |
| Real yearly trophic carrying capacity correlated by representation within the local fish assemblages | 121,912 |

**Table 7.** Ecological carrying capacity for *Romanichthys valsanicola* in its native areal, the Valsan River.

| | | Number of Supported Individuals |
|---|---|---|
| Behavioral carrying capacity | Minimum density (maximum territory size) | 8241 |
| | Average density (average territory size) | 14,542 |
| | Maximum density (minimum territory size) | 61,804 |
| Trophic carrying capacity | | 121,912 |

A remarkable finding is that actually neither the available habitat size nor the food availability are the limiting population size parameters for *Romanichthys valsanicola*, since all estimations produced considerably higher potential individual numbers than even the most optimistic assessments concerning the sculpin perch effectives up to date, which provide population sizes of about 100 [28,33]. One important clue is given by the fact that the trophic carrying capacity of the natural habitat is extremely high, and it could easily support the requirements of an increasing population, even at minimum territory size and high individual densities, as long as shelter is provided by conserving the rocky structure of the riverbed and restricting the boulder extraction illegal activities. The present study represents a useful conceptual approach for further studies that intend to estimate critical ecological parameters required for a realistic approach to biodiversity conservation initiatives. A carrying capacity value offers the background for complex population viability estimates and populational trend assessments that respond to the actual specific problematics. In our opinion, the biology and populational ecology of the sculpin perch should be more thoroughly investigated in terms of population size and structure investigations by mark-recapture studies carried out over longer periods to outline the seasonal and diachronic variations. Behavioral studies also play an important role, and their findings might solve current gaps in data concerning feeding seasonality and food production in the ecosystem. Another issue of critical importance is represented by the study of the local fish assemblages and their variation over time to document in a reliable manner the percent of total biomass occupied by the species of conservation concern.

## 4. Conclusions

Both trophic availability as well as populational density present encouragingly high values for the sculpin perch in the Valsan River, indicating that the natural habitat could host a considerably higher number of *Romanichthys valsanicola*, on one hand, and the fact that neither food nor territory are the critical limiting factors for the species 'populational decline.

This aspect, represented by the high food and spatial availability that is well beyond the ecological carrying capacity of the ecosystem, as presented by the authors, suggests that further conservation efforts such as ex situ, captive reproduction of the sculpin perch for repopulation actions should be highly endorsed. Such actions would contribute to the development of an ex situ stock of individuals that may be used in repopulation initiatives within the actual natural habitat to increase the wild stocks of the species from the remnant populational core area represented by the Valsan River with a high chance that the species will naturally populate the adjacent streams of its historical distribution, namely, the Arges River and the Doamnei River.

Further investigation efforts should concentrate on realistic field investigations for population size (number of individuals) and structure (sex ratio, percentual contribution of age classes, generation size calculations, age-related mortality) assessment, involving estimations that use several different indicators (Lincoln–Peterson with the Chapman and Bailey modifications, Schnabel, Cormack–Jolly–Seber [49–55]) to provide a less biased interpretation on the sculpin perch population size within its natural habitat and to set the informational foundation required for complex evaluations, such as population viability analyses (Vortex 10 [56]). Fine-scale bathymetric investigations are advised to test the correlation between populational estimates and underwater relief and large stones/boulders density, hydrodynamic parameters and other issues that may be associated with the specific preferences for feeding and reproduction. Another important direction for future investigations is proper optimal habitat mapping and spatial modeling processes that would reflect the specific ecological preferences of *Romanichthys valsanicola* within the Valsan River as well as a proper percentual quantification of the species' preferred habitat within the whole riverine ecosystem, which should also include estimates of ecological and trophic interspecific niche sharing in terms of food competition with other species.

**Author Contributions:** Conceptualization, L.B. and G.D.; Data curation, L.B., G.D. and I.S.; Formal analysis, L.B., G.D., M.B., M.R., I.S. and A.J.; Funding acquisition, G.D., M.B. and E.H.; Investigation, L.B., M.R., I.S. and A.J.; Methodology, L.B., G.D. and M.R.; Project administration, G.D., M.B. and E.H.; Resources, L.B., M.R., E.H. and I.S.; Supervision, G.D. and M.B.; Validation, L.B. and G.D.; Visualization, L.B., M.B. and I.S.; Writing—original draft, L.B., G.D., E.H. and I.S.; Writing—review and editing, L.B., G.D., M.B., M.R., E.H., I.S. and A.J. All authors have read and agreed to the published version of the manuscript.

**Funding:** This research received no external funding.

**Institutional Review Board Statement:** Capturing of individuals was carried out under the authorization for scientific fishing activities No. 10/14.02.2022 (from 14.02.2022 to 31.12.2022), issued by the National Agency for Fishing and Aquaculture, within the Ministry of Agriculture and Rural Development.

**Data Availability Statement:** The data presented in this study are available on request from the corresponding author. The data are not publicly available due to privacy and ethical aspects, concerning the critically endangered status of the species and its endemism, and represents sensitive information that cannot be made public except within the scientific community involved in the conservation of the species. Non-sensitive data is contained within the article as maps and tables.

**Acknowledgments:** This work was carried out through the project "Institutional development of the National Institute for Research and Development in Environmental Protection Bucharest in order to increase the capacity and performance in the field of environmental protection and climate change (2022–2024)- No.39PFE/30.12.2021". The field research activities were carried out within the project: Conservation status assessment for the fish species of community interest on the national scale and identification of the favorable/unfavorable conservation status, according to the 17th

article of the Habitat Directive 92/43/CEE, regarding the country report for Romania, financed through the Large Infrastructure Operational Program (POIM), Priority Axis 4—Environmental protection through biodiversity conservation measures, air quality monitoring and decontamination of historically polluted sites, Specific Objective 4.1 "Increasing the degree of biodiversity protection and conservation and reconstruction of degraded ecosystems".

**Conflicts of Interest:** The authors declare no conflict of interest.

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
