# Peer review of "Understanding the Ecosystem Carrying Capacity for Romanichthys valsanicola, a Critically Endangered Freshwater Fish Endemic to Romania, with Considerations upon Trophic Offer and Behavioral Density"

_diversity, doi:10.3390/d15060748_

Round 1
Reviewer 1 Report
Manuscript entitled "Understanding the ecosystem carrying capacity for Romanichthys valsanicola M. Dumitrescu, P. Bănărescu & N. Stoica, 1957, (PISCES: PERCIFORMES: PERCIDAE), a critically endangered freshwater fish endemic to Romania, with considerations upon trophic offer and behavioral density" is an interesting text on a valuable, endangered species of ichthyofauna.
General comments - the text fits the DIVERSITY profile and after taking into account the comments it can be published.
Detailed notes:
The title is way too long. Suggest to shorten - M. Dumitrescu, P. Bănărescu & N. Stoica, 1957, (PISCES: PERCIFORMES: PERCIDAE) is not necessary, because the text mentions it.
Too many Keywords – standard there should be max 5-7 - it is necessary to process them, eg IUCN, fish, etc.
1. Introduction - the chapter is a bit unfinished. It would be good to give some information about the species itself - if it was discovered 66 years ago, it is a young species among the ichthyofauna. Where are its natural habitats? What threats does it currently have and why is it threatened with extinction? Are there any efforts to reintroduce Romanichthys valsanicola? I suggest expanding the chapter by a few sentences and adding a few references.
2. Materials and Methods
Section 2.1 Specific area contains too much redundant information that should be at the end of the manuscript - Conservation status assessment for the fish species of community interest on the national scale and identification of the favorable/unfavorable conservation
status, according to the 17th article of the Habitat Directive 92/43/CEE, regarding the country report for Romania, financed through the Large Infrastructure Operational
Program (POIM), Priority Axis 4 – Environmental protection through biodiversity conservation measures, air quality monitoring and decontamination of historically polluted sites, Specific Objective 4.1 “Increasing the degree of biodiversity protection and conservation and reconstruction of degraded ecosystems”.
I understand that the Authors must and want to post it, but not in MM.
Figure 2 - unreadable, where is the Historical area? There is a grate in the legend, but there is no grate in the picture.
Discussion - practically non-existent. I understand that the Authors didn't have much literature to discuss, so I suggest combining this chapter with the results to form one: Results and Discussion
Author Response
The required revisions were made.

Reviewer 2 Report
Title: Understanding the ecosystem carrying capacity for Romanichthys valsanicola M. Dumitrescu, P. Bănărescu & N. Stoica, 1957, (PISCES: PERCIFORMES: PERCIDAE), a critically endangered freshwater fish endemic to Romania, with considerations upon trophic offer and behavioral density
Introduction
Line 34 – insert reference: Freyhof, J. & Kottelat, M. 2008. Romanichthys valsanicola. The IUCN Red List of Threatened Species 2008: e.T19740A900820
Line 44 – remove ‘study’
Line 51 – add ‘a’ after throughout
Line 58 – do you know that no harm was caused to the fish? Possibly remove this part.
Line 59 – what measurements were taken?
Line 71 – Valsan River (and elsewhere)
Line 86 – The only existing peer reviewed publication dealing ….
Line 133 – 10 to 10
Line 156 – difference or differentiate
Line 163 - argues instead of arguments
Line 171 – individuals (typo)
Line 200 – transverses ‘past’ the stomach quickly? Not sure what this means
Line 244 – chosing to choosing
Line 259 – applicable?
Line 278 – delete ‘base’
Some incorrect use of words - see comments
Author Response
Line 34 – insert reference: Freyhof, J. & Kottelat, M. 2008. Romanichthys valsanicola. The IUCN Red List of Threatened Species 2008: e.T19740A900820 - the reference is already inserted at line 35; If you prefer so, i also duplicated it at line 34 as well;
Line 44 – remove ‘study’ - i could not find the word "study" at line 44, but considering the context, I can presume you referred to the same word at line 43. I removed it.
Line 51 – add ‘a’ after throughout - done;
Line 58 – do you know that no harm was caused to the fish? Possibly remove this part. I know for sure that no harm was dealt to the captured individuals since all fishes were caught by electronarcosis, measured and weighed on site and released immediately. Since you prefer removing the section, I did so.
Line 59 – what measurements were taken? - total and standard length, weight (to be used in further studies regarding the body mass index and the Fulton condition factor; I added a phrase refering to the measurements and references to total and standard length;
Line 71 – Valsan River (and elsewhere) - I have corrected that according to your sugestion;
Line 86 – The only existing peer reviewed publication dealing …. - modified;
Line 133 – 10 to 10 - modified;
Line 156 – difference or differentiate - modified;
Line 163 - argues instead of arguments - modified;
Line 171 – individuals (typo) - modified, thank you!
Line 200 – transverses ‘past’ the stomach quickly? Not sure what this means - the phrase was replaced with ", indicating thus that the digestion processes may have a fast rate for this species, hence we can assume that the fullness index value represents the daily food intake per individual.".
Line 244 – chosing to choosing - modified, thank you!
Line 259 – applicable? - "a conceptual approach that is appliable for further studies" was replaced with " - represents a useful conceptual approach for further studies"
Line 278 – delete ‘base’ - the phrase was replaced as follows: "Further conservation efforts should concentrate on realistic field investigations for population size assessment, involving several estimators "